# SEENN: Towards Temporal Spiking Early-Exit Neural Networks

**Yuhang Li**
Yale University
New Haven, CT, USA
yuhang.li@yale.edu

**Tamar Geller**
Yale University
New Haven, CT, USA
tamar.geller@yale.edu

**Youngeun Kim**
Yale University
New Haven, CT, USA
youngeun.kim@yale.edu

**Priyadarshini Panda**
Yale University
New Haven, CT, USA
priya.panda@yale.edu

## Abstract

Spiking Neural Networks (SNNs) have recently become more popular as a biologically plausible substitute for traditional Artificial Neural Networks (ANNs). SNNs are cost-efficient and deployment-friendly because they process input in both spatial and temporal manner using binary spikes. However, we observe that the information capacity in SNNs is affected by the number of timesteps, leading to an accuracy-efficiency tradeoff. In this work, we study a fine-grained adjustment of the number of timesteps in SNNs. Specifically, we treat the number of timesteps as a variable conditioned on different input samples to reduce redundant timesteps for certain data. We call our method **S**piking **E**arly-**E**xit **N**eural **N**etworks (**SEENNs**). To determine the appropriate number of timesteps, we propose SEENN-I which uses a confidence score thresholding to filter out the uncertain predictions, and SEENN-II which determines the number of timesteps by reinforcement learning. Moreover, we demonstrate that SEENN is compatible with both the directly trained SNN and the ANN-SNN conversion. By dynamically adjusting the number of timesteps, our SEENN achieves a remarkable reduction in the average number of timesteps during inference. For example, our SEENN-II ResNet-19 can achieve **96.1**% accuracy with an average of **1.08** timesteps on the CIFAR-10 test dataset. Code is shared at https://github.com/Intelligent-Computing-Lab-Yale/SEENN.

## 1 Introduction

Deep learning has revolutionized a range of computational tasks such as computer vision and natural language processing [1] using Artificial Neural Networks (ANNs). These successes, however, have come at the cost of tremendous computational demands and high latency [2]. In recent years, Spiking Neural Networks (SNNs) have gained traction as an energy-efficient alternative to ANNs [3, 4, 5]. 0SNNs infer inputs across a number of timesteps as opposed to ANNs, which infer over what is essentially a single timestep. Moreover, during each timestep, the neuron in an SNN either fires a spike or remains silent, thus making the output of the SNN neuron binary and sparse. Such spike-based computing produces calculations that substitute multiplications with additions.

In the field of SNN research, there are two main approaches to getting an SNN: (1) directly training SNNs from scratch and (2) converting ANNs to SNNs. Direct training seeks to optimize an SNN using methods such as spike timing-based plasticity [6] or surrogate gradient-based optimization [7, 8]. In contrast, the ANN-SNN conversion approach [9, 10, 11, 12, 13, 14] uses the feature representation

37th Conference on Neural Information Processing Systems (NeurIPS 2023).

of a pre-trained ANN and aims to replicate it in the corresponding SNN. Both methods have the potential to achieve high-performance SNNs when implemented correctly.

Despite the different approaches, both training-based and conversion-based SNNs are limited by binary activations. As a result, the key factor that affects their information processing capacity is the **number of timesteps**. Expanding the number of timesteps enables SNNs to capture more features in the temporal dimension, which can improve their accuracy in conversion and training. However, a larger number of timesteps increases latency and computational requirements, resulting in a lower acceleration ratio, and yielding a tradeoff between accuracy and time. Therefore, current efforts to enhance SNN accuracy often involve finding ways to achieve it with fewer number of timesteps.

In this paper, we propose a novel approach to improve the tradeoff between accuracy and time in SNNs. Specifically, our method allows each input sample to have a varying number of timesteps during inference, increasing the number of timesteps only when the current sample is hard to classify, resulting in an early exit in the time dimension. We refer to this approach as **S**piking **E**arly-**E**xit **N**eural **N**etworks (**SEENNs**). To determine the optimal number of timesteps for each sample, we propose two methods: SEENN-I, which uses confidence score thresholding to output a confident prediction as fast as possible; and SEENN-II, which employs reinforcement learning to find the optimal policy for the number of timesteps. Our results show that SEENNs can be applied to both conversion-based and direct training-based approaches, achieving new state-of-the-art performance for SNNs. In summary, our contributions are threefold:

1. We introduce a new direction to optimize SNN performance by treating the number of timesteps as a variable conditioned to input samples.
2. We propose Spiking Early-Exit Neural Networks (SEENNs) and use two methods to determine which timestep to exit: confidence score thresholding and reinforcement learning optimized with policy gradients.
3. We evaluate our SEENNs on both conversion-based and training-based models with large-scale datasets like CIFAR and ImageNet. For example, our SEENNs can use $\sim$**1.1** timesteps to achieve similar performance with a 6-timestep model on the CIFAR-10 dataset.

## 2    Related Work

### 2.1    Spiking Neural Networks

**ANN-SNN Conversion**  Converting ANNs to SNNs utilizes the knowledge from pre-trained ANNs and replaces the ReLU activation in ANNs with a spike activation in SNNs [10, 12, 15, 16]. The conversion-based method, therefore, seeks to match the features in two different models. For example, [10, 15] studies how to select the firing threshold to cover all the features in an ANN. [17] studies using a smaller threshold and [13] proposes to use a bias shift to better match the activation. Based on the error analysis, [14] utilizes a parameter calibration technique and [18] further changes the training scheme in ANN.

**Direct Training of SNN**  Direct training from scratch allows SNNs to operate within extremely few timesteps. In recent years, the number of timesteps used to train SNNs has been reduced from more than 100 [19, 20] to less than 5 [21]. The major success is based on the spatial-temporal backpropagation [7, 22] and surrogate gradient estimation of the firing function [23]. Through gradient-based learning, recent works [24, 25, 26, 27, 28, 29] propose to optimize not only parameters but also firing threshold and leaky factor. Moreover, loss function [30], surrogate gradient estimation [31], batch normalization [32], activation distribution [33], membrane potential normalization/regularization [34, 35] are also factors that affect the learning behavior in direct training and have been investigated properly. Our method, instead, focuses on the time dimension, which is fully complementary to both conversion and training.

### 2.2    Conditional Computing

Conditional computing models can boost the representation power by adapting the model architectures, parameters, or activations to different input samples [36]. BranchyNet [37] and Conditional Deep Learning [38] add multiple classifiers in different layers to apply spatial early exit to ANNs. SkipNet [39] and BlockDrop [40] use a dynamic computation graph by skipping different blocks

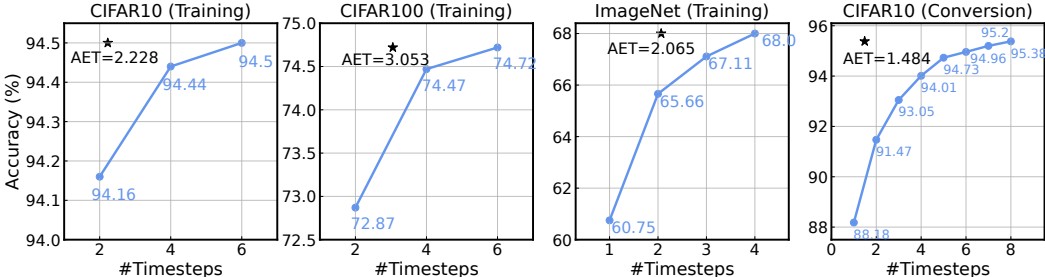

**Figure 1:** The accuracy of spiking ResNet under different numbers of timesteps on CIFAR10, CIFAR100, and ImageNet datasets, either by direct training [30] or by conversion [18].

based on different samples. CondConv [41] and Dynamic Convolution [36] use the attention mechanism to change the weight in convolutional layers according to input features. Squeeze-and-Excitation Networks [42] proposes to reweight different activation channels based on the global context of the input samples. To the best of our knowledge, our work is the first to incorporate conditional computing into SNNs.

## 3 Methodology

In this section, we describe our overall methodology and algorithm details. We start by introducing the background on fixed-timestep spiking neural networks and then, based on a strong rationale, we introduce our SEENN method.

### 3.1 Spiking Neural Networks

SNNs simulate biological neurons with Leaky Integrate-and-Fire (LIF) layers [43]. In each timestep, the input current in the $\ell$-th layer charges the membrane potential $\mathbf{u}$ in the LIF neurons. When the membrane potential exceeds a threshold, a spike $\mathbf{s}$ will fire to the next layer, as given by:

$$\mathbf{u}^\ell[t+1] = \tau\mathbf{u}^\ell[t] + \boldsymbol{W}^\ell\mathbf{s}^{\ell-1}[t], \tag{1}$$

$$\mathbf{s}^\ell[t+1] = H(\mathbf{u}^\ell[t+1] - V), \tag{2}$$

where $\tau \in (0,1]$ is the leaky factor, mimicking natural potential decay. $H(\cdot)$ is the Heaviside step function and $V$ is the firing threshold. If a spike fires, the membrane potential will be reset to 0: $(\mathbf{u}[t+1] = \mathbf{u}[t+1]*(1-\mathbf{s}[t+1]))$. Following existing baselines, in direct training, we use $\tau = 0.5$ while in conversion we use $\tau = 1.0$, transforming to the Integrate-and-Fire (IF) model. Moreover, the reset in conversion is done by subtraction: $(\mathbf{u}[t+1] = \mathbf{u}[t+1] - V_{th}\mathbf{s}[t+1])$ as suggested by [12]. Now, denote the overall spiking neural network as a function $f_T(\mathbf{x})$, its forward propagation can be formulated as

$$f_T(\mathbf{x}) = \frac{1}{T}\sum_{t=1}^{T} h \circ g^L \circ g^{L-1} \circ g^{L-2} \circ \cdots g^1(\mathbf{x}), \tag{3}$$

where $h(\cdot)$ denotes the final linear classifier, and $g^\ell(\cdot)$ denotes the $\ell$-th block of backbone networks. $L$ represents the total number of blocks in the network. A block contains a convolutional layer to compute input current ($\boldsymbol{W}^\ell\mathbf{s}^{\ell-1}$), a normalization layer [21], and a LIF layer. In this work, we use a direct encoding method, *i.e.* using $g^1(\mathbf{x})$ to encode the input tensor into spike trains, as done in recent SNN works [24, 30, 44]. In SNNs, we repeat the inference process for $T$ times and average the output from the classifier to produce the final result.

### 3.2 Introducing Early-Exit to SNNs

Conventionally, the number of timesteps $T$ is set as a fixed hyper-parameter, causing a tradeoff between accuracy and efficiency. Here, we also provide several $accuracy - T$ curves of spiking ResNets in Fig. 1. These models are trained from scratch or converted to 4, 6, or 8 timesteps and we evaluate the performance with the available numbers of timesteps. For the CIFAR-10 dataset,

increasing the number of timesteps from 2 to 6 only brings 0.34% top-1 accuracy gain, at the cost of 300% more latency. *In other words, the majority of correct predictions can be inferred with much fewer timesteps.*

The observation in Fig. 1 motivates us to explore a more fine-grained adjustment of $T$. We are interested in an SNN that can adjust $T$ based on the characteristics of different input images. Hypothetically, each image is linked to an *difficulty* factor, and this SNN can identify this difficulty to decide how many timesteps should be used, thus eliminating unnecessary timesteps for easy images. We refer to such a model as a spiking early-exit neural network (SEENN).

To demonstrate the potential of SEENN, we propose a metric that calculates the minimum timesteps needed to perform correct prediction averaged on the test dataset, *i.e.* the lowest timesteps we can achieve without comprising the accuracy, and it is based on the following assumption:

**Assumption 3.1.** Given a spiking neural network $f_T$, if it can correctly predict $\mathbf{x}$ with $t$ timesteps, then it always outputs correct prediction for any $t'$ such that $t \leq t' \leq T$.

This assumption indicates the inclusive property of different timesteps. In Appendix A, we provide empirical evidence to support this assumption. Let $\mathbb{C}_t$ be the set of correct predicted input samples for timestep $t$, we have $\mathbb{C}_1 \subseteq \mathbb{C}_2 \subseteq \cdots \subseteq \mathbb{C}_T$ based on Assumption 3.1. Also denote $\mathbb{W} = \overline{\mathbb{C}_T}$ as the wrong prediction set, we propose the *averaged earliest timestep (AET)* metric, given by

$$\text{AET} = \frac{1}{N}\left(|\mathbb{C}_1| + T|\mathbb{W}| + \sum_{t=2}^{T} t(|\mathbb{C}_t| - |\mathbb{C}_{t-1}|)\right), \tag{4}$$

where $|\cdot|$ returns the cardinal number of the set, and $|\mathbb{C}_t| - |\mathbb{C}_{t-1}|$ returns the number of samples that belong to $\mathbb{C}_t$ yet not to $\mathbb{C}_{t-1}$. $N = |\mathbb{C}_T| + |\mathbb{W}|$ is the total number of samples in the validation dataset. The AET metric describes an ideal scenario where correct predictions are always inferred using the minimum number of timesteps required, while still preserving the original accuracy. It's worth noting that incorrect samples are inferred using the maximum number of timesteps, as it is usually not possible to determine if a sample cannot be correctly classified before inference.

In Fig. 1, we report the AET in each case. For models directly trained on CIFAR10 or CIFAR100, the AET remains slightly higher than 2 (note that the minimum number of timesteps is 2). With merely 11% more latency added to the 2-timestep SNN on CIFAR10 ($T = 2.228$), we can achieve an accuracy equal to a 6-timestep SNN. The converted SNNs also only need a few extra time steps. This suggests the huge potential for using early exit in SNNs.

Despite the potential performance boost from the early exit, it is impossible to achieve the AET effect in practice since we cannot access the label in the test set. Therefore, the question of how to design an efficient and effective predictor that determines the number of timesteps for each input is non-trivial. In the following sections, we propose two methods, SEENN-I and SEENN-II, to address this challenge.

### 3.3 SEENN-I

In SEENN-I, we adopt a proxy signal to determine the difficulty of the input sample—confidence score. Formally, let the network prediction probability distribution be $\mathbf{p} = \text{softmax}(f_t(\mathbf{x})) = [p_1, p_2, \ldots, p_M]$, where $M$ is the number of object classes, the confidence score (CS) is defined as

$$\text{CS} = \max(\mathbf{p}), \tag{5}$$

which means the maximum probability in $\mathbf{p}$. The CS is a signal that measures the level of uncertainty. If the CS is high enough (*e.g.* CS= 0.99), the prediction distribution will be highly deterministic; otherwise (*e.g.* CS= $\frac{1}{M}$), the prediction distribution is uniform and extremely uncertain.

A line of work [37, 45] has shown that the level of uncertainty is highly correlated with the accuracy of the neural networks. For input samples with deterministic output prediction, the neural network typically achieves high accuracy, which corresponds to relatively *easy* samples. Hence, we can utilize this property as an indicator of how many time steps should be assigned to this sample. We adopt a simple thresholding mechanism, *i.e.* given a preset threshold $\alpha$, we iterate each timestep and once the confidence score is higher than $\alpha$, the accumulated output is used for prediction. Appendix B demonstrates the distribution of confidence scores changes with different numbers of timesteps. A diagram describing this method can be found in Fig. 2(a).

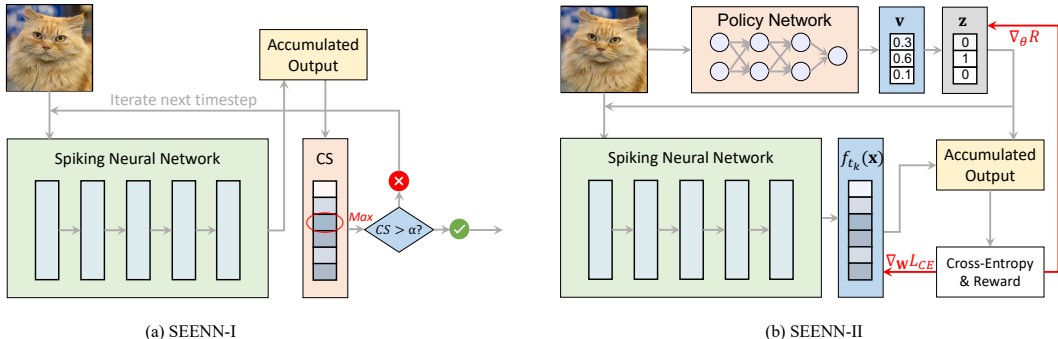

(a) SEENN-I                        (b) SEENN-II

**Figure 2:** The frameworks of our proposed SEENNs. (a): SEENN-I uses the confidence score thresholding for determining the optimal number of timesteps, (b): SEENN-II leverages a policy network to predict the number of timesteps, optimized by reinforcement learning.

## 3.4 SEENN-II

Our SEENN-I is a post-training method that can be applied to an off-the-shelf SNN. It is easy to use, but the SNN is not explicitly trained with the early exit, so the full potential of early exit is not exploited through SEENN-I. In this section, we propose an "early-exit-aware training" method for SNNs called SEENN-II.

SEENN-II is built from reinforcement learning to directly predict the difficulty of the image. Specifically, we define an action space $\mathcal{T} = \{t_1, t_2, \cdots, t_n\}$, which contains the candidates for the number of timesteps that can be applied to an input sample $\mathbf{x}$. To determine the optimal timestep candidates, we develop a policy network that generates an $n$-dimensional *policy vector* to sample actions. During training, a reward function is calculated based on the policy and the prediction result, which is generated by running the SNN with the number of timesteps suggested by the policy vector. Unlike traditional reinforcement learning [46], our problem does not consider state transition as the policy can predict all actions at once. We also provide a diagram describing the overall method in Fig. 2(b).

Formally, consider an input sample $\mathbf{x}$ and a policy network $f_p$ with parameter $\theta$, we define the policy of selecting the timestep candidates as an $n$-dim categorical distribution:

$$\mathbf{v} = \mathrm{softmax}(f_p(\mathbf{x}; \theta)), \quad \pi_\theta(\mathbf{z}|\mathbf{x}) = \prod_{k=1}^{n} \mathbf{v}_k^{\mathbf{z}_k}, \tag{6}$$

where $\mathbf{v}$ is the probability of the categorical distribution, obtained by inferring the policy networks with a $\mathrm{softmax}$ function. Thus $\mathbf{v}_k$ represents the probability of choosing $t_k$ as the number of timesteps in the SNN. An action $\mathbf{z} \in \{0, 1\}^n$ is sampled based on the policy $\mathbf{v}$. Here, $\mathbf{z}$ is a one-hot vector since only one timestep can be selected. Note that the policy network architecture is made sufficiently small such that the cost of inferring the policy is negligible compared to SNN (see architecture details in Sec. 4.1).

Once we obtain an action vector $\mathbf{z}$, we can evaluate the prediction using the target number of timesteps, *i.e.* $f_t(\mathbf{x})$. Our objective is to minimize the number of timesteps we used while not sacrificing accuracy. Therefore, we associate the actions taken with the following reward function:

$$R(\mathbf{z}) = \begin{cases} \frac{1}{2^{t_k|_{\mathbf{z}_k=1}}} & \text{if correct prediction} \\ -\beta & \text{if incorrect prediction} \end{cases}. \tag{7}$$

Here, $t_k|_{\mathbf{z}_k=1}$ represents the number of timesteps selected by $\mathbf{z}$. Here, the reward function is determined by whether the prediction is correct or incorrect. If the prediction is correct, then we incentivize early exit by assigning a larger reward to a policy that uses fewer timesteps. However, if the prediction is wrong, we penalize the reward with $\beta$, which serves the role to balance the accuracy and the efficiency. As an example, a large $\beta$ leads to more correct predictions but also more timesteps.

**Gradient Calculation in the Policy Network** To this end, our objective for training the policy network is to maximize the expected reward function, given by:

$$\max_\theta \; \mathbb{E}_{\mathbf{z} \sim \pi_\theta}[R(\mathbf{z})]. \tag{8}$$

In order to calculate the gradient of the above objective, we utilize the policy gradient method [46] to compute the derivative of the reward function w.r.t. $\theta$, given by

$$\nabla_\theta \mathbb{E}[R(\mathbf{z})] = \mathbb{E}[R(\mathbf{z})\nabla_\theta \log \pi_\theta(\mathbf{z}|\mathbf{x})] = \mathbb{E}[R(\mathbf{z})\nabla_\theta \log \prod_{k=1}^{n} \mathbf{v}_k^{\mathbf{z}_k}] = \mathbb{E}[R(\mathbf{z})\nabla_\theta \sum_{k=1}^{n} \mathbf{z}_k \log \mathbf{v}_k]. \quad (9)$$

Moreover, unlike other reinforcement learning which relies on Monte-Carlo sampling to compute the expectation, our method can compute the exact expectation. During the forward propagation of the SNN, we can store the intermediate accumulated output at each $t_k$, and calculate the reward function using the stored accumulated output $f_{t_k}(\mathbf{x})$. Since $\pi_\theta(\mathbf{z}|\mathbf{x})$ is a categorical distribution, we can rewrite Eq. (9) as

$$\nabla_\theta \mathbb{E}[R(\mathbf{z})] = \sum_{k=1}^{n} R(\mathbf{z}|_{\mathbf{z}_k=1})\mathbf{v}_k\nabla_\theta \log \mathbf{v}_k, \quad (10)$$

where $R(\mathbf{z}|_{\mathbf{z}_k=1})$ is the reward function evaluated with the output prediction using $t_k$ timesteps.

### 3.5 Training SEENN

In this section, we describe the training methods for our SEENN. To train the model for SEENN-I, we explicitly add a cross-entropy function to each timestep, thus making the prediction in early timesteps better, given by

$$\min_{\mathbf{W}} \frac{1}{n} \sum_{k=1}^{n} L_{CE}(f_{t_k}(\mathbf{x}), \mathbf{W}, \mathbf{y}), \quad (11)$$

where $L_{CE}$ denotes the cross-entropy loss function and $\mathbf{y}$ is the label vector. This training objective is essentially Temporal Efficient Training (TET) loss proposed in [30]. We find this function can enhance the performance from every $t_k$ compared to $L_{CE}$ applied to only the maximum number of timesteps. As for conversion-based SEENN-I, we do not modify any training function. Instead, we directly apply the confidence score thresholding to the converted model.

To train the SEENN-II, we first employ TET loss to train the model for several epochs without involving the policy network. This can avoid the low training accuracy in the early stage of the training which may damage the optimization of the policy network. Then, we jointly optimize the SNN and the policy network by

$$\min_{\mathbf{W},\theta} \mathbb{E}_{\mathbf{z}\sim\pi_\theta}[-R(\mathbf{z}) + L_{CE}(f_{t_{k|\mathbf{z}_k=1}}(\mathbf{x}), \mathbf{W}, \mathbf{y})]. \quad (12)$$

Note that we do not train SEENN-II for the converted model since in conversion, a pre-trained ANN is used to obtain the converted SNN and does not warrant any training.

## 4 Experiments

To demonstrate the efficacy and the efficiency of our SEENN, we conduct experiments on popular image recognition datasets, including CIFAR10, CIFAR100 [47], ImageNet [48], and an event-stream dataset CIFAR10-DVS [49]. Moreover, to show the compatibility of our method, we compare SEENN with both training-based and conversion-based state-of-the-art methods. Finally, we also provide some hardware and qualitative evaluation on SEENN.

### 4.1 Implementation Details

In order to implement SEENN-I, we adopt the code framework of TET [30] and QCFS [18], both of which provide open-source implementation. All models are trained with a stochastic gradient descent optimizer with a momentum of 0.9 for 300 epochs. The learning rate is 0.1 and decayed following a cosine annealing schedule [50]. The weight decay is set to $5e-4$. For ANN pre-training in QCFS, we set the step $l$ to 4. We use Cutout [51] and AutoAugment [52] for better accuracy as adopted in [31, 32].

To implement SEENN-II, we first take the checkpoint of pre-trained SNN in SEENN-I and initialize the policy network. Then, we jointly train the policy network as well as the SNN. The policy network we take for the CIFAR dataset is ResNet-8, which contains only 0.547% computation of ResNet-19.

**Table 1:** Accuracy-$T$ comparison of **direct training** SNN methods on CIFAR datasets.

| Method | Model | CIFAR-10 | | | CIFAR-100 | | |
|---|---|---|---|---|---|---|---|
| | | $Acc_1(T_1)$ | $Acc_2(T_2)$ | $Acc_3(T_3)$ | $Acc_1(T_1)$ | $Acc_2(T_2)$ | $Acc_3(T_3)$ |
| ResNet-20 | Dspike [31] | 93.13 (2) | 93.16 (4) | 94.25 (6) | 71.28 (2) | 73.35 (4) | 74.24 (6) |
| | tdBN [21] | 92.34 (2) | 92.92 (4) | 93.16 (6) | - | - | - |
| | TET [30] | 94.16 (2) | 94.44 (4) | 94.50 (6) | 72.87 (2) | 74.47 (4) | 74.72 (6) |
| | RecDis-SNN [33] | 93.64 (2) | 95.53 (4) | 95.55 (6) | - | 74.10 (4) | - |
| ResNet-19 | IM-Loss [53] | 93.85 (2) | 95.40 (4) | 95.49 (6) | - | - | - |
| | TEBN [32] | 95.45 (2) | 95.58 (4) | 95.60 (6) | 78.07 (2) | 78.71 (4) | 78.76 (6) |
| | **SEENN-I (Ours)** | **96.07** (1.09) | **96.38** (1.20) | **96.44** (1.34) | **79.56** (1.19) | **81.42** (1.55) | **81.65** (1.74) |
| | **SEENN-II (Ours)** | **96.01** (1.08) | - | - | **80.23** (1.21) | - | - |
| VGG-16 | Diet-SNN [25] | - | - | 92.70 (5) | - | - | 69.67 (5) |
| | Temporal prune [54] | 93.05 (**1**) | 93.72 (2) | 93.85 (3) | 63.30 (**1**) | 64.86 (2) | 65.16 (3) |
| | **SEENN-I (Ours)** | **94.07** (1.08) | **94.41** (1.20) | **94.56** (1.46) | **71.87** (1.12) | **73.53** (1.46) | **74.30** (1.84) |
| | **SEENN-II (Ours)** | **94.36** (1.09) | - | - | **72.76** (1.15) | - | - |

**Table 2:** Accuracy-$T$ comparison of **ANN-SNN conversion** SNN methods on CIFAR datasets.

| Method | Model | CIFAR-10 | | | CIFAR-100 | | |
|---|---|---|---|---|---|---|---|
| | | $Acc_1(T_1)$ | $Acc_2(T_2)$ | $Acc_3(T_3)$ | $Acc_1(T_1)$ | $Acc_2(T_2)$ | $Acc_3(T_3)$ |
| ResNet-20 | Opt. [13] | 92.41 (16) | 93.30 (32) | 93.55 (64) | 63.73 (16) | 68.40 (32) | 69.27 (64) |
| | Calibration [14] | 94.78 (32) | 95.30 (64) | 95.42 (128) | - | - | - |
| | OPI [30] | 75.44 (8) | 90.43 (16) | 94.82 (32) | 23.09 (8) | 52.34 (16) | 67.18 (32) |
| ResNet-18 | QCFS [18] | 75.44 (2) | 90.43 (4) | 94.82 (8) | 19.96 (2) | 34.14 (4) | 55.37 (8) |
| | **SEENN-I (Ours)** | **91.08 (1.18)** | **93.63 (1.40)** | **95.08 (2.01)** | **39.33 (2.57)** | **56.99 (4.41)** | **65.48 (6.19)** |

We also provide the details of architecture in Appendix E and measure the latency/energy of the policy network in Sec. 4.3. For the ImageNet dataset, we downsample the image resolution to $112 \times 112$ for the policy network. Both the policy network and the SNN are finetuned for 75 epochs using a learning rate of 0.01. Other training hyper-parameters are kept the same as SEENN-I.

## 4.2 Comparison to SOTA work

**CIFAR Training** We first provide the results on the CIFAR-10 and CIFAR-100 datasets [47]. We test the architectures in the ResNet family [55] and VGG-16 [56]. We summarize both the direct training comparison as well as the ANN-SNN conversion comparison in Table 1 and Table 2, respectively. Since the $T$ in our SEENN can be variable based on the input, we report the average number of timesteps in the test dataset. As we can see from Table 1, existing direct training methods usually use 2, 4, and 6 timesteps in inference. However, increasing the number of timesteps from 2 to 6 brings marginal improvement in accuracy, for example, 95.45% to 95.60% in TEBN [32]. Our SEENN-I can achieve **96.07**% with only **1.09** average timesteps, which is **5.5**× lower than the state of the art. SEENN-II gets a similar performance with SEENN-I in the case of CIFAR-10, but it can achieve 0.7% higher accuracy than SEENN-I on the CIFAR-100 dataset.

**Comparison with 1-Timestep SNN** Reducing the number of timestep to 1 is another way towards low latency SNN [54, 57]. However, this method ignores the sample-wise difference and may lead to a larger accuracy drop. In Table 1, we compare VGG-16 with even 1 timestep [54]. Our SEENN-I achieves 8.5% accuracy improvement at the cost of 12% more latency on the CIFAR-100 dataset.

**CIFAR Conversion** We also include the results of ANN-SNN conversion using SEENN-I in Table 2. Here, the best existing work is QCFS [18], which can convert the SNN in lower than 8 timesteps. We directly run SEENN-I with different choices of confidence score threshold $\alpha$. Surprisingly, on the CIFAR-10 dataset, our SEENN-I can convert the model with **1.4** timesteps, and get **93.63**% accuracy. Instead, the QCFS only gets 75.44% accuracy using 2 timesteps for all input images. By selecting a higher threshold, we can obtain **95.08**% accuracy with **2.01** timesteps, which uses **4**× lower number of timesteps than QCFS under the same accuracy level.

**Table 3:** Accuracy-$T$ comparison on ImageNet dataset.

| Model | Method | T | Acc. | Model | Method | T | Acc. |
|-------|--------|---|------|-------|--------|---|------|
| | *Direct Training of SNNs* | | | | *ANN-SNN Conversion* | | |
| ResNet-34 | tdBN [21] | 6 | 63.72 | ResNet-34 | Opt [13] | 32 | 33.01 |
| | TET [30] | 6 | **64.79** | | | 64 | 59.52 |
| | TEBN [32] | 6 | 64.29 | | Calibration [14] | 32 | 64.54 |
| | **SEENN-I (Ours)** | 2.28 | 63.65 | | | 64 | 71.12 |
| | | 3.38 | 64.66 | | QCFS [18] | 16 | 59.35 |
| | **SEENN-II (Ours)** | 2.40 | 64.18 | | | 32 | 69.37 |
| SEW-ResNet-34 | SEW [58] | 4 | 67.04 | | | 64 | 72.35 |
| | TET [30] | 4 | 68.00 | | **SEENN-I (Ours)** | 23.47 | 70.18 |
| | TEBN [32] | 4 | **68.28** | | | 29.53 | 71.84 |
| | **SEENN-I (Ours)** | 1.66 | 66.21 | | | | |
| | | 2.35 | 67.99 | | | | |
| | **SEENN-II (Ours)** | 1.79 | 67.48 | | | | |

**ImageNet** We compare the direct training and conversion methods on the ImageNet dataset. The results are sorted Table 3. For direct training, we use two baseline networks: vanilla ResNet-34 [55] and SEW-ResNet-34 [58]. For SEENN-I, we directly use the pre-trained checkpoint from TET [30]. It can be observed that our SEENN-I can achieve the same accuracy using only 50% of original number of timesteps. For example, SEENN-I ResNet-34 reaches the maximum accuracy in **2.35** timesteps. SEENN-II can obtain a similar performance by using a smaller $T$, *i.e.* 1.79. For conversion experiments, we again compare against QCFS using ResNet-34. Our SEENN-I obtains **70.2%** accuracy with **23.5** timesteps, higher than a 32-timestep QCFS model.

**CIFAR10-DVS** Here, we compare our SEENN-I on an event-stream dataset, CIFAR10-DVS [49]. Following existing baselines, we train the SNN with a fixed number of 10 timesteps. From Table 4 we can find that SEENN-I can surpass all existing work except TET with only **2.53** timesteps, amounting to nearly **4×** faster inference. Moreover, SEENN-II can get an accuracy of **82.6** using 4.5 timesteps.

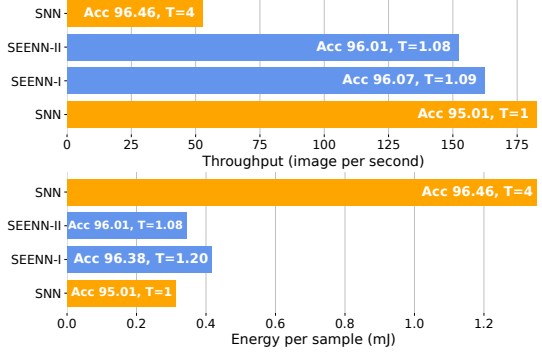

**Figure 3:** Comparison of latency (throughput) and energy consumption between SNN and SEENN.

| Model | Method | T | Acc. |
|-------|--------|---|------|
| | *Direct Training of SNNs* | | |
| ResNet-19 | tdBN [21] | 10 | 67.8 |
| VGGSNN | PLIF [24] | 20 | 74.8 |
| ResNet-18 | Dspike [31] | 10 | 75.4 |
| ResNet-19 | RecDis-SNN [33] | 10 | 72.4 |
| VGGSNN | TET [30] | 10 | **83.1** |
| VGG-SNN | **SEENN-I (Ours)** | 2.53 | 77.6 |
| | | 5.17 | 82.7 |
| | **SEENN-II (Ours)** | 4.49 | 82.6 |

**Table 4:** Accuracy-$T$ comparison on CIFAR10-DVS dataset.

## 4.3 Hardware Efficiency

In this section, we analyze the hardware efficiency of our method, including latency and energy consumption. Due to the sequential processing nature of SNNs, the latency is generally proportional to $T$ on most hardware devices. Therefore, we directly use a GPU (NVIDIA Tesla V100) to evaluate the latency (or throughput) of SEENN. For energy estimation, we follow a rough measure to count only the energy of operations that is adopted in previous work [18, 25, 31], as SNNs are usually deployed on memory-cheap devices (detailed in Appendix D). Fig. 3 plots the comparison of inference throughput and energy. It can be found that our SEENN-I simultaneously improves inference speed while reducing energy costs. Meanwhile, the policy network in SEENN-II only brings marginal effect and does not impact the overall inference speed and energy cost, demonstrating the efficiency of our proposed method. We show ImageNet results in Appendix C.

## 4.4 Ablation Study

In this section, we conduct the ablation study on our SEENN. In particular, we allow SEENN to flexibly change their balance between the accuracy and the number of timesteps. For example, SEENN-I can adjust the confidence score threshold $\alpha$ and SEENN-II can adjust the penalty value $\beta$ defined in the reward function. To demonstrate the impact of different hyper-parameters selection, we utilize SEENN-I evaluated with different $\alpha$. Fig. 4 shows the comparison. The yellow line denotes the trained SNN with a fixed number of timesteps while the blue line denotes the corresponding SEENN-I with 6 different thresholds. We test the SEW-ResNet-34 on the ImageNet dataset and the ResNet-19 on the CIFAR10 dataset. For both datasets, our SEENN-I has a higher $accuracy - T$ curve than the vanilla SNN, which confirms that **our SEENN improves the accuracy-efficiency tradeoff**. Moreover, our SEENN-I largely reduces the distance between the AET coordinate and the $accuracy - T$ curve, meaning that our method is approaching the upper limit of the early exit. On the right side of the Fig. 4, we additionally draw the composition pie charts of SEENN-I, which shows how many percentages of inputs are using 1, 2, 3, or 4 timesteps, respectively. It can be shown that, as we gradually adjust $\alpha$ from 0.4 to 0.9 for the SEW-ResNet-34, the percentage of inputs using 1 timestep decreases (73.4% to 30.7%). For the CIFAR10 dataset, we find images have a higher priority in using the first timestep, ranging from 95.4% to 56.2%.

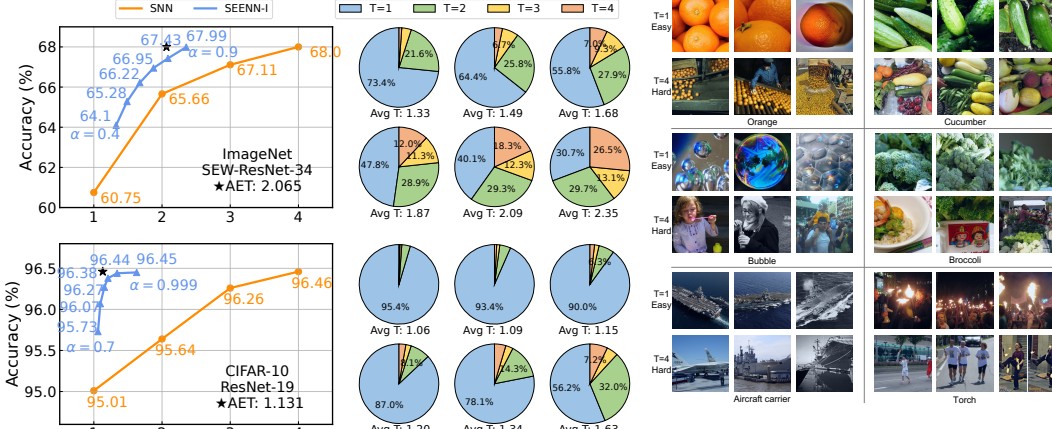

**Figure 4:** *Left:* Accuracy vs. the number of timesteps curve. *Right:* Pie charts indicating the composition of input images using different numbers of timesteps for inference.

**Figure 5:** Qualitative assessment using input images from ImageNet dataset. We select images from *orange, cucumber, bubble, broccoli, aircraft carrier, and torch* classes and separate them according to SEENN.

## 4.5 Qualitative Assessment

In this section, we conduct a qualitative assessment of SEENN by visualizing the input images that are separated by our SEENN-II, or the policy network. Specifically, we take the policy network and let it output the number of timesteps for each image in the ImageNet validation dataset. In principle, the policy network can differentiate whether the image is *easy* or *hard* so that easy images can be inferred with less number of timesteps and hard images can be inferred with more number of timesteps. Fig. 5 provides some examples of this experiment, where images are chosen from orange, cucumber, bubble, broccoli, aircraft carrier, and torch classes in the ImageNet validation dataset. We can find that $T = 1$ (easy) images and $T = 4$ (hard) images have huge visual discrepancies. As an example, the orange, cucumber, and broccoli images from $T = 1$ row are indeed easier to be identified, as they contain single objects in a clean background. However, in the case of $T = 4$, there are many irrelevant objects overlapped with the target object, or there could be many small samples of target objects which makes it harder to identify. For instance, in the cucumber case, there are other vegetables that increase the difficulty of identifying them as cucumbers. These results confirm our hypothesis that visually simpler images are indeed easier and can be correctly predicted using a fewer number of timesteps.

## 5 Conclusion

In this paper, we introduce SEENN, a novel attempt to allow a varying number of timesteps on an input-dependent basis. Our SEENN includes both a post-training approach (confidence score thresholding) and an early-exit-aware training approach (reinforcement learning for selecting the appropriate number of timesteps). Our experimental results show that SEENN is able to find a sweet spot that maintains accuracy while improving efficiency. Moreover, we show that the number of timesteps selected by SEENNs is related to the visual difficulty of the image. By taking an input-by-input approach during inference, SEENN is able to achieve state-of-the-art accuracy with less computational resources.

## Acknowledgment

This work was supported in part by CoCoSys, a JUMP2.0 center sponsored by DARPA and SRC, the National Science Foundation (CAREER Award, Grant #2312366, Grant #2318152), TII (Abu Dhabi), and the DoE MMICC center SEA-CROGS (Award #DE-SC0023198).

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

# A  Verification of Assumption 3.1

In Assumption 3.1, we conjecture that if a correct prediction is made by $f_t(\mathbf{x})$, then for all $t' \geq t$ that $f_{t'}(\mathbf{x})$ is also correct. And based on this assumption, we propose the equation to compute AET, *i.e.* Eq. (4). Here, to validate if our assumption holds, we propose another metric: *empirical AET*, given by

$$\widetilde{\text{AET}} = \frac{1}{N} \left( \sum_{i=1}^{N} \tilde{t}_i \right), \tag{13}$$

where $\tilde{t}_i$ is the actual earliest number of timesteps that predicts the correct class for the $i$-th sample. Note that, similar to AET, if the network cannot make correct predictions in any timesteps, then we set $\tilde{t}_i$ to the maximum number of timesteps.

Consider an example that does not satisfy Assumption 3.1, for example, the prediction results for the first 4 timesteps are {False, True, False, True}. The empirical AET will obtain 2 while the AET will obtain $2 - 3 + 4 = 3$. Therefore, by comparing the difference between the AET and the empirical AET we can verify if Assumption 3.1 holds. Table 5, as shown below, demonstrates the AET and the empirical AET comparison across different models and datasets. We can find that difference is very small, often less than 0.05. The only dataset that creates a slightly larger difference is CIFAR10-DVS. Indeed, the event-stream dataset may provide more variations over time. Nevertheless, the potential for early exit is still high.

**Table 5:** Comparison between ATE and empirical AET across models and datasets.

| Model | Dataset | $\max T$ | AET | $\widetilde{\text{AET}}$ |
|---|---|---|---|---|
| *Direct Training of SNNs* | | | | |
| ResNet-19 | CIFAR-10 | 4 | 1.1309 | 1.1188 |
| ResNet-19 | CIFAR-100 | 4 | 1.6075 | 1.5616 |
| ResNet-34 | ImageNet | 6 | 2.9026 | 2.7781 |
| SEW-ResNet-34 | ImageNet | 4 | 2.0646 | 2.0092 |
| VGGSNN | CIFAR10-DVS | 10 | 3.096 | 2.774 |

# B  Distribution of Confidence Scores

In Fig. 2, we demonstrate the distribution of confidence scores of ResNet-19 on the CIFAR-10 validation dataset. Obviously, with a higher number of timesteps more input data have higher confidence scores. This aligns with our target to filter out easy input data.

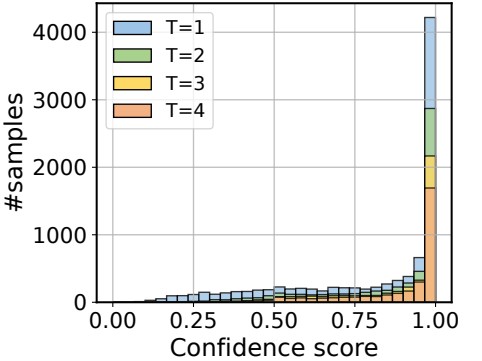 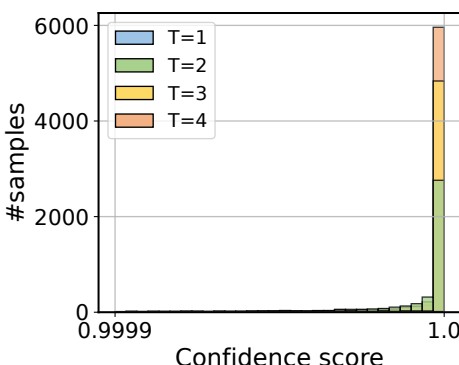

**Figure 6:** The distribution of confidence scores of ResNet-19 on CIFAR-100, left: $[0, 0.9999]$, right: $[0.9999, 1]$

## C  Hardware Efficiency on ImageNet

We demonstrate the results in Fig. 7.

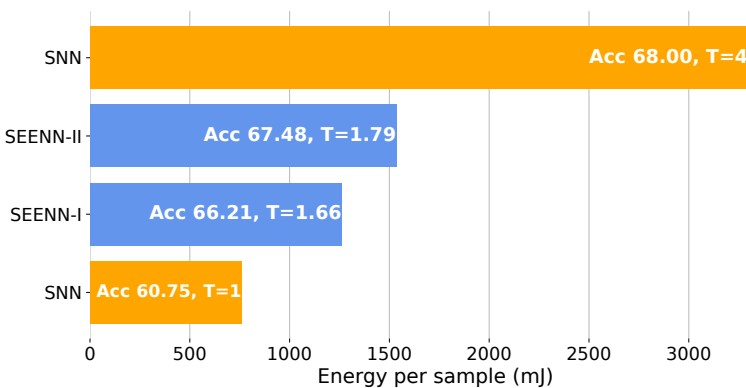

**Figure 7:** Hardware efficiency on ImageNet dataset.

## D  Hardware Evaluation

For latency measurement, we directly tested on GPU with the Pytorch framework. The latency measurement is computed across the whole test dataset. For example, the throughput calculation is given by

$$\text{Throughput} = \frac{\text{Test set inference time}}{\text{Number of test samples}}. \tag{14}$$

For energy estimation, we adopt the conventional way to measure the addition and multiplication operations for the entire SNN inference. The addition costs $0.9pJ$ and the multiplication costs $4.6pJ$, respectively.

## E  Architecture Details

The major network architecture we adopt in this work is ResNet-series architecture. There are many variants of ResNets used in the field of SNNs, *e.g.* ResNet-18, ResNet-19, and ResNet-20. They are also mixedly referred to in existing literature [18, 21, 30, 55]. Therefore, to avoid confusion in these ResNet architectures, in this section, we sort out the details of the network configurations.

Table 6 summarizes the different ResNets we used in our CIFAR experiments. For ImageNet models, we use the standard ResNet-34 [55] and SEW-ResNet-34 [58] which are well-defined. The basic difference between the four ResNets is the channel configurations and the block configurations. Therefore, their actual FLOPs difference is a lot larger than the difference suggested by their names. For example, our policy network (ResNet-8) only contains $0.547\%$ number of operations of ResNet-19. Therefore, the cost of running SEENN-II is almost negligible.

**Table 6:** The architecture details of ResNets. * denotes that the first residual block contains downsample layer to reduce the feature resolution.

|  | **ResNet-8** | **ResNet-18** |
|---|---|---|
| conv1 | $3 \times 3, 16, s1$ | $3 \times 3, 64, s1$ |
| stage1 | $\begin{pmatrix} 3 \times 3, 16 \\ 3 \times 3, 16 \end{pmatrix} \times 1$ | $\begin{pmatrix} 3 \times 3, 64 \\ 3 \times 3, 64 \end{pmatrix} \times 2$ |
| stage2 | $\begin{pmatrix} 3 \times 3, 32 \\ 3 \times 3, 32 \end{pmatrix}^* \times 1$ | $\begin{pmatrix} 3 \times 3, 128 \\ 3 \times 3, 128 \end{pmatrix}^* \times 2$ |
| stage3 | $\begin{pmatrix} 3 \times 3, 64 \\ 3 \times 3, 64 \end{pmatrix}^* \times 1$ | $\begin{pmatrix} 3 \times 3, 256 \\ 3 \times 3, 256 \end{pmatrix}^* \times 2$ |
| stage4 | N/A | $\begin{pmatrix} 3 \times 3, 512 \\ 3 \times 3, 512 \end{pmatrix}^* \times 2$ |
| pooling | Global average pooling | |
| classifier | $(64, O)$ FC | $(512, O)$ FC |
|  | **ResNet-19** | **ResNet-20** |
| conv1 | $3 \times 3, 128, s1$ | $3 \times 3, 16, s1$ |
| stage1 | $\begin{pmatrix} 3 \times 3, 128 \\ 3 \times 3, 128 \end{pmatrix} \times 3$ | $\begin{pmatrix} 3 \times 3, 16 \\ 3 \times 3, 16 \end{pmatrix} \times 2$ |
| stage2 | $\begin{pmatrix} 3 \times 3, 256 \\ 3 \times 3, 256 \end{pmatrix}^* \times 3$ | $\begin{pmatrix} 3 \times 3, 32 \\ 3 \times 3, 32 \end{pmatrix}^* \times 2$ |
| stage3 | $\begin{pmatrix} 3 \times 3, 512 \\ 3 \times 3, 512 \end{pmatrix}^* \times 2$ | $\begin{pmatrix} 3 \times 3, 64 \\ 3 \times 3, 64 \end{pmatrix}^* \times 2$ |
| pooling | Global average pooling | |
| classifier | $(512, O)$ FC | $(64, O)$ FC |

