# OpenReview forum: "SEENN: Towards Temporal Spiking Early Exit Neural Networks"
_NeurIPS.cc/2023/Conference — NeurIPS 2023 poster_

### Official Review · Reviewer_Sp8z · 2023-07-04

**Soundness:** 2 fair
**Presentation:** 3 good
**Contribution:** 2 fair
**Rating:** 5
**Confidence:** 4

**Summary:**

This paper considers to adaptively determine inference time steps of spiking neural networks to improve the tradeoff between accuracy and time. Two methods are proposed. The first one uses confidence score thresholding. The second one introduces an additional policy network to predict the number of timesteps by reinforcement learning. Experiments show the effectiveness of the proposed methods for direct SNN training and ANN-SNN conversion methods.

**Strengths:**

1. This paper considers early-exit of spiking neural networks, which can effectively improve the efficiency of SNNs. The idea of dynamically determining inference time steps based on the difficulty of inputs is interesting.

2. Extensive experiments on static and neuromorphic datasets as well as qualitative assessment are conducted.


**Weaknesses:**

1. Dynamic inference time steps of SNNs, especially based on confidence scores, were also explored by (possibly concurrent) recent works [1,2], which can be discussed. The idea of confidence score is very simple and straight-forward.

2. For SEENN-II, an additional policy network is required for inference, and it is unclear how this can be deployed. From the descriptions (and codes), the policy network is an ANN rather than SNN, so it is not compatible with the main SNN for hardware deployment. This poses challenges and also arouses questions of why considering such hybrid architectures. Additionally, SEENN-II seems not to be flexible, i.e. after training it cannot make tradeoff between accuracy and time/energy.

3. From the descriptions, it is not clear enough whether the energy consumption estimation of SEENN-II consider policy networks. This should be included and more detailedly discussed. For example, on ImageNet, is the policy network an ANN ResNet-34 (as shown in code)? Then it may consume more than the SNN part. The energy result on ImageNet is missing.

[1] Wu, D., Jin, G., Yu, H., Yi, X., & Huang, X. (2023). Optimising Event-Driven Spiking Neural Network with Regularisation and Cutoff. arXiv preprint arXiv:2301.09522.

[2] Li, C., Jones, E., & Furber, S. (2023). Unleashing the Potential of Spiking Neural Networks by Dynamic Confidence. arXiv preprint arXiv:2303.10276.


**Questions:**

Can the authors discuss more about conceptually how to determine the inference of (deep) SNNs (in small time steps) with confidence signals from the top layer? Note that the common time steps of SNNs simply refer to time steps for each layer and do not consider propagation across layers. However, if we consider asynchronous neurons, e.g. on neuromorphic hardware, it also takes time to propagate signals across layers, and the actual delay may be layer number plus time steps per layer. For deep SNNs (e.g. >18 layers in this paper), it takes much time to obtain the confidence score from the top layer, which may be longer than the small time steps per layer. Then how can it effectively control such small time steps?

**Limitations:**

The authors do not discuss limitations and societal impact.

---

> ### Author Rebuttal · Authors · 2023-08-09
>
> Thank you for your efforts in reviewing our article and providing constructive feedback. We’d like to reply to your concerns in detail.
>
> Q1: Dynamic inference time steps of SNNs were also explored by (possibly concurrent) recent works [1,2], which can be discussed.
>
> A1: Thank you for pointing out the concurrent works [1] and [2]. We were unaware of these papers during our manuscript preparation, and we appreciate the opportunity to clarify the distinctions between our work and theirs.
>
> While [1] and [2] concentrate on converted SNNs with early exit using post-training methods, our SEENN approach is applied to both conversion and direct training of SNNs. Their methods rely on metrics like spike differences and dynamic confidence scores, whereas we utilize confidence score metrics. Beyond the post-training approach, we introduce a training-aware method (SEENN-II) that leverages reinforcement learning to optimize early exit performance. This innovation enhances accuracy within the same number of timesteps. Our work provides a comprehensive empirical comparison with existing SNN literature, encompassing both conversion and direct training methodologies.
>
> We acknowledge the relevance of [1] and [2] and will incorporate references to these papers in our revised manuscript. This inclusion will enrich the context and highlight the unique contributions of our SEENN approach.
>
> Q2: For SEENN-II, an additional policy network is required for inference, and it is unclear how this can be deployed. Additionally, SEENN-II seems not to be flexible.
>
> A2: Thanks for the question. Choosing the policy network architecture is open to discussion. Technically, any neural network would suffice this role. Indeed, if we use two SNNs it might have better hardware compatibility. Then, the question is whether we should use the early exit on policy network architecture and so on. We have avoided this recursive paradox and used an extremely small ANN (ResNet-8 with much fewer channels), which only occupies 0.5% of the FLOPs of the ResNet-19.
>
> Meanwhile, we would also like to point out existing work that tries to involve both ANNs and SNNs to enhance the representation ability and efficiency. It is possible to use two types of neural networks on the edge devices collaboratively, e.g. [3].
>
> As for the tradeoff problem, we agree with this notice. Here, we want to clarify that a training-aware approach for SNNs has always been fixed to certain timesteps. To be more specific, in the direct training of SNNs, all existing works report the test accuracy when using the same $T$ in training. Even though they can test under lower timesteps, their reported accuracy on the different numbers of timesteps always uses different networks trained with the corresponding numbers of timesteps. We argue that this is a common problem for all training-aware approaches. Post-training algorithms, like our SEENN-I and conversion methods, can flexibly trade off between accuracy and time. In conclusion, our SEENN-I and SEENN-II align with the existing SNNs method in terms of practicality.
>
>
> Q3: From the descriptions, it is not clear enough whether the energy consumption estimation of SEENN-II consider policy networks. The energy result on ImageNet is missing.
>
> A3: We included all costs from policy networks and in line 277 we described that “Meanwhile, the policy network in SEENN-II only brings marginal effect and does not impact the overall inference speed and energy cost”. This is due to the extremely efficient design of the policy network architecture.
>
> To measure the energy, we used the same energy per operation times the number of operations as in [27], where the ANN policy network architecture uses multiplications and additions as well. In our code, we use ResNet-8 with an 8x fewer channel size (when compared to ResNet-19) for the policy network, occupying 0.5% FLOPs of the ResNet-19. Hence, the actual cost induced by the policy network has been made as minuscule as possible. Here, we report the energy results on the ImageNet dataset in the rebuttal PDF file.
>
> Q4: Can the authors discuss more about conceptually how to determine the inference of (deep) SNNs (in small time steps) with confidence signals from the top layer?  How can it effectively control such small time steps?
>
> A4: This is a good question. To the best of our understanding, the reviewer is asking about an asynchronous pipeline inference implementation, where later timesteps start getting processed before the current timestep reaches the top layer and causes unnecessary energy waste.
>
> First, we would like to clarify that not all hardware devices implement this technique, since it requires lots of area to map the full network to the hardware. For example, Loihi2 [4] only has 8192 neurons in one core, which is not enough for an 18-layer network.
>
> Second, if we have to use SEENN on this type of hardware. We can avoid unnecessary energy waste with several methods. (1) Instead of sending the next timestep of current input to the pipeline, we can send the next input data. The next timestep of the current input will be processed only when the confidence score is computed. We put a figure in the rebuttal PDF to show this scheme.
> (2) Determine the confidence score at early layers. This can be done by training a classifier in the early layers (similar to ANN early exit) and using the signal from early layers to determine the exit timesteps. (3) Predict timesteps before the inference as we did in SEENN-II.
>
> Q5: The authors do not discuss limitations and societal impact.
>
> A5: We apologize for not mentioning the limitations and broader impact. Please check our general response.
>
> Reference:
>
> [3] Zhao, Rong, et al. "A framework for the general design and computation of hybrid neural networks." Nature communications 13.1 (2022): 3427.
> [4] Davies, Mike. "Taking neuromorphic computing to the next level with Loihi2." Intel Labs’ Loihi 2 (2021): 1-7.

---

> > ### Comment · Reviewer_Sp8z · 2023-08-15
> >
> > I would like to thank the authors for their detailed responses. Most of my questions are addressed and I will raise my score. Some additional comments: 1. Directly trained SNNs can be tested under different timesteps for trade-off and some previous works reported such accuracy [1,2]; 2. The design to send the next input data to the pipeline requires frequent memory exchange and additional memory (e.g., for membrane potentials), which can cause much additional energy consumption.
> >
> > [1] Temporal efficient training of spiking neural network via gradient re-weighting. ICLR 2022.
> >
> > [2] Online training through time for spiking neural networks. NeurIPS 2022.

---

> > > ### Author Response · Authors · 2023-08-18
> > > **Authors reply**
> > >
> > > We'd like to thank reviewer Sp8z for his appreciation and genuine discussion of our work. We'd like to discuss the two comments raised by reviewer Sp8z.
> > >
> > > (1) *Directly trained SNNs can be tested under different timesteps for trade-off and some previous works reported such accuracy [1,2]*
> > >
> > > Reply: Yes, technically directly trained SNNs can be tested with other numbers of timesteps. Nonetheless, we emphasize that these two works report did not report those accuracies when they are compared against other paper. For example, TET[1] has shown its ability to adjust $T$ in Figure 4, but the model trained with $T=4$ only gets 70.5% accuracy evaluated at $T=2$. However, they report 72.87% accuracy in their main table (Table 3). OTTT[2] also has a huge accuracy gap (7%) between $T=2$ and $T=6$, and they did not compare $T=2$ results with their current state-of-the-art methods like tdBN, TET.
> > >
> > > Having said that, our emphasis is that the results we presented in Table 1 are evaluated under a fair setting with existing methods.
> > >
> > > (2) *The design to send the next input data to the pipeline requires frequent memory exchange and additional memory (e.g., for membrane potentials), which can cause much additional energy consumption.*
> > >
> > > Reply: We agree with the reviewer on this matter. Let's formulate this entire question from the beginning. In such a pipeline inference architecture, we can describe the overall latency as
> > >
> > > $$Latency(T) = C + \Delta (T-1),$$
> > >
> > > where $C$ is a constant representing the latency to finish all layer's computation and $\Delta$ is the latency interval between any two timesteps. As the reviewer suggested, if we have a very deep network (i.e. $C$ is large) under a low number of timesteps (i.e., max $T$ is small, for example 4), SEENN-I cannot save much energy. This is true.
> > >
> > > However, the problem is under such cases, any timesteps reduction is not meaningful since $Latency(4)-Latency(1)$ has only $3\Delta$ magnitude, which is a lot lower than $C$. One may just use the full timesteps to utilize the full performance of SNNs.
> > >
> > > If the max $T$ is a lot higher, for example, $>500$, which is often the case for this type of hardware, timesteps reduction can bring a significant energy reduction. The ratio of energy saving is
> > >
> > > $$Ratio = \frac{C + \Delta (\hat{T}-1)}{C+\Delta(T-1)}$$
> > >
> > > where $\hat{T}$ is the average number of timesteps for all test samples in SEENN-I.
> > >
> > > To summarize, the relationship between $C$, $\Delta$, and max $T$ will impact the overall saving ratio.
> > > The choice of $C$ is a hardware design problem and decides the upper limit of the acceleration ratio in SEENN-I or other timestep reduction work. In a few extreme cases ($C \gg \Delta \times T$), timestep reduction brings a limited advantage. This hardware design problem is somewhat beyond the scope of our work as it impacts the common algorithmic aspects in SNNs like network architecture and the choice of $T$.

---

### Official Review · Reviewer_DV4t · 2023-07-05

**Soundness:** 4 excellent
**Presentation:** 4 excellent
**Contribution:** 4 excellent
**Rating:** 7
**Confidence:** 5

**Summary:**

This paper proposes an accuracy-efficiency tradeoff method by adjusting the number of timesteps in SNNs, which is new and interesting. The authors accomplish this idea with two methods that uses a confidence score thresholding and reinforcement learning. These methods can be applied to directly training SNN and the ANN-SNN both, and results show these methods can save energy greatly but with negligible accuracy decreasing. I like the work that has potential and would provide a new direction for the following SNN work.

**Strengths:**

1. well-written, easy to read.
2.The idea to adjust the number of timesteps to balance accuracy-efficiency is new and interesting.
3. The method is simple and easy to follow.
4. Experimental results are really good. The method can keep similar performance while reduce cost.

**Weaknesses:**

To show the effectiveness of the method. A similar timesteps for other methods should be provided, for example, 2 or 3 timesteps for TET on ImageNet.

**Questions:**

1.Can this method be used to other backbones, like transformer?
2.What is the performance of the method using vanilla CE loss?

**Limitations:**

1.I find no potential negative societal impact.

---

> ### Author Rebuttal · Authors · 2023-08-09
>
> We appreciate your positive feedback on our work. Please check our response to your questions and concerns.
>
> Q1: To show the effectiveness of the method. A similar timesteps for other methods should be provided, for example, 2 or 3 timesteps for TET on ImageNet.
>
> A1: Thanks for the suggestion. We report the TET SEW-ResNet-34 (denoted as Static SNN) accuracy from T=1 to T=4 on the ImageNet dataset as well as our SEENN accuracy. The results are shown in the table below.
>
> | Method     | T     | Accuracy |
> |------------|-------|----------|
> | Static SNN | 1     | 60.78    |
> | Static SNN | 2     | 65.74    |
> | Static SNN | 3     | 67.11    |
> | Static SNN | 4     | 68.00    |
> | SEENN-I    | 1.66  | 66.20    |
> | SEENN-I    | 2.35  | 67.99    |
> | SEENN-II   | 1.79  | 67.48    |
>
> Q2: Can this method be used to other backbones, like transformer?
>
> A2: Yes, our method can be seamlessly integrated into any SNN backbones with any number of timesteps. For example, we train a SpikeFormer [1] on the CIFAR10 dataset and demonstrate the improvement from our SEENN-I in the following table.
>
> | SpikeFormer |       | SEENN SpikFormer |       |
> |-------------|-------|------------------|-------|
> | T=1         | 89.39 | T=1.23           | 93.47 |
> | T=2         | 93.96 | T=1.35           | 93.91 |
> | T=3         | 94.30 | T=1.82           | 94.28 |
> | T=4         | 94.51 | T=2.11           | 94.49 |
>
> Q3: What is the performance of the method using vanilla CE loss?
>
> A3: This is a good question. When using vanilla CE loss, the performance of early timesteps will inevitably decrease, thus damaging the early exit performance. We refer you to our reply to Reviewer ZThW where we show the SEENN performance on a pretrained SNASNet checkpoint, which is trained with vanilla CE loss function. It can be found that the acceleration of SEENN is still effective. However, the reduction in timesteps is relatively smaller, presumably brought by a higher AET value (i.e., Eq. 4 in the main manuscript).

---

### Official Review · Reviewer_J7UY · 2023-07-05

**Soundness:** 3 good
**Presentation:** 3 good
**Contribution:** 3 good
**Rating:** 7
**Confidence:** 5

**Summary:**

This paper introduces a new inference scheme for spiking neural networks (SNNs), the early exit on the time dimension. To make sure that relatively easy images can be predicted with less number of timesteps, this work build two frameworks to identify the appropriate timestep to minimize the latency while maintaining decent performance. The first method is an on-the-fly approach and the second one is more complicated that requires finetuning.  The authors test their SEENN on various recognition benchmarks and obtain quite good accuracy with even lower number of timesteps compared to existing papers.

**Strengths:**

1. Compared to other SNNs papers that focus on either conversion and training, what is work studied is a universal approach (i.e. time) that can be applied to any types SNNs.
2. The methodology is technically sound and covers different user resources (whether finetune or just plug-and-play)
3. Experiments results is thorough and solid and verify the effectiveness of this approach.

**Weaknesses:**

1. There is no empirical evidence showing that the confidence score will increase along with the number of timesteps. The authors are suggested to add a figure to show the confidence score evolution.

The above limitations has been explained in the rebuttal.

**Questions:**

Can SEENN-II be applied to the conversion approach? if not, why is that, please elaborate it.

I suggest the following response about this problem is discussed into the revision.

**Limitations:**

see above.

---

> ### Author Rebuttal · Authors · 2023-08-09
>
> We appreciate your positive feedback on our work. Please check our response to your questions and concerns.
>
> Q1: There is no empirical evidence showing that the confidence score will increase along with the number of timesteps. The authors are suggested to add a figure to show the confidence score evolution.
>
> A1: Thanks for your suggestion. We agree with the reviewer that a visualization of the confidence score would help the readers understand our SEENN-I mechanism. Here, we draw the confidence score distribution on test images. We split the range into [0, 0.9999] and [0.9999, 1.0], otherwise the figure would distort too much. We put the figure into the rebuttal PDF file, please refer to it there.
>
> It can be clearly observed that the number of test samples moving to [0.9999, 1.0]  has increased if we increase the number of timesteps. This means that the network prediction is getting more confident as we increase the number of timesteps.
>
> Moreover, we also measure the mean/variance of confidence scores over the test batches, as shown below. We can find the mean/variance continues to increase/decrease as we increase the number of timesteps.
>
> | Timesteps   | 1     | 2     | 3     | 4     |
> |-------------|-------|-------|-------|-------|
> | CS Mean     | 0.816 | 0.923 | 0.953 | 0.964 |
> | CS Variance | 0.059 | 0.024 | 0.014 | 0.010 |
>
> Q2: Can SEENN-II be applied to the conversion approach? if not, why is that, please elaborate it.
>
> A2: Thanks for this question. In SEENN-II, we jointly optimize the policy network and the SNNs. If we apply SEENN-II to converted SNNs, we can focus on training the policy network only and keep the converted SNNs frozen.
>
> However, many computation resources and the whole training data are required to train the policy network. This setup breaks the assumption that ANN-SNN conversion is done when there are limited computation resources and training data. Therefore, we did not include the SEENN-II to ANN-SNN conversion methods in our initial draft because we think the comparison is unfair.
>
> To demonstrate that SEENN-II can work effectively with the converted model, we’d like to provide a result for QCFS-based (Bu et al., 2022b) converted ResNet-18 on the CIFAR-10 dataset. We train a policy network and the average predicted number of timesteps is 1.35, achieving 94.43% accuracy. The improvement is consistent with the results shown in our paper.

---

> > ### Comment · Reviewer_J7UY · 2023-08-12
> >
> > Thanks for the responses, my concerns have been resovled.
> >
> > I appreciate the additional experiment and the discussion in the above response, the corresponding content should be added into the revision if accepted.

---

### Official Review · Reviewer_ZThW · 2023-07-24

**Soundness:** 2 fair
**Presentation:** 2 fair
**Contribution:** 2 fair
**Rating:** 6
**Confidence:** 3

**Summary:**

This paper proposes a novel manner to improve the efficiency of the spiking neural network. Specifically, the SEENN is proposed to determine the appropriate number of timesteps, which therefore reduce the inference time-cost. The proposed method is evaluated on CIFAR-10 and Imagenet, achieving good performance.

**Strengths:**

+ The main contribution of this paper is to treat the number of timesteps as a variable in SNN model. Accordingly, several variations of early-exit manner is deigned for better accuracy-efficiency tradeoff.
+ The paper writing and organization is good.

**Weaknesses:**

- The proposed method for determining the best timesteps is not novel. The first manner is to simply set the right number with the confidence score, while the second one introduces a policy network for better prediction. Although above two manners could work, these would inevitably introduce extra computation cost or human-based prior, which is not expected. In addition, the proposed SNN model still rely heavily on the ANN backbones, e.g., Resnet or VGGnet.
- As for the hardware efficiency, the adopted nvidia V 100 maybe not proper, which is not designed for SNN application.
- The ablation study is not clear, it seems the SEEN adopts different backbones with SNN.

**Questions:**

See weakness.

**Limitations:**

None.

---

> ### Author Rebuttal · Authors · 2023-08-09
>
> Thank you for your efforts in reviewing our article and providing constructive feedback. We’d like to reply to your concerns in detail.
>
> Q1: The proposed method for determining the best timesteps is not novel. The first manner is to simply set the right number with the confidence score, while the second one introduces a policy network for better prediction. Although above two manners could work, these would inevitably introduce extra computation cost or human-based prior, which is not expected. In addition, the proposed SNN model still rely heavily on the ANN backbones, e.g., Resnet or VGGnet.
>
> A1: Thanks for your question. Our proposed methodology aims to minimize the extra computation cost of SEENN. For SEENN-I, calculating the confidence score is quite fast, only involving a softmax and a max function to determine the score, which is negligible compared to the whole network inference (Eq. 5). For SEENN-II, we use an extremely tiny network (ResNet-8) with 8x fewer channels than ResNet-19 (See details in our appendix C), which only occupies 0.5% FLOPs of the ResNet-19. In our experiments, we compared the energy/latency with static SNN baselines and found that SEENN incurs much lower hardware costs even with these extra computations.
>
> As for the human prior problem, we should clarify that this is the same problem with **all** SNNs works. Under the static time steps setting, it is required to pre-set the number of timesteps for the SNN before the inference. As such, the human prior is needed to choose more efficiency or more accuracy by changing the number of timesteps. In our SEENN, this is similar to setting the $\alpha$ or $\beta$ to balance efficiency and accuracy.
>
> For the architecture concern, we’d like to emphasize that our work adjusts the timesteps of any SNN, agnostic to the SNN architecture used. For example, we can use the SNAS-Net [1] on the CIFAR10, a work that searches for unique SNN network architecture. We apply our early exit mechanism to it. The results are shown below:
>
> | SNASNet |       | SEENN SNASNet  |       |
> |---------|-------|----------------|-------|
> | T=1     | 73.69 | T=1.39         | 84.59 |
> | T=2     | 84.79 | T=1.99         | 90.85 |
> | T=3     | 90.87 | T = 2.36       | 92.21 |
> | T=4     | 92.66 | T = 2.76       | 93.05 |
> | T=5     | 93.39 | T = 3.14       | 93.37 |
>
> In addition to NAS-based networks, we refer you to our reply to Reviewer DV4t where we provide the results on the Transformer based network architecture.
>
> Q2: As for the hardware efficiency, the adopted nvidia V100 maybe not proper, which is not designed for SNN application.
>
> A2: Thanks for your question. It is true that Nvidia GPUs are not optimized for SNNs. Therefore, most SNN acceleration work cannot be tested on GPUs. However, in this paper, we show that reducing the number of timesteps of SNNs can lead to acceleration on any inference platform, including GPUs that are not even optimized for SNNs. Our SEENN can be inherently accelerated on other platforms like neuromorphic hardware and in-memory computing (IMC) architectures. Here, we show the implementation of SEENN on a publicly available IMC architecture simulator [2]. The experiments use ResNet-19 architecture on the CIFAR10 dataset. Besides the accuracy metric, we use the relative Energy-Delay-Product (EDP) with respect to Static SNN at 4 timesteps as the hardware metric:
>
> The results are shown in the table below. It can be found that our SEENN also shares a significant amount of acceleration on other hardware platforms.
>
> | Method     | T    | Accuracy | Relative EDP (%) |
> |------------|------|----------|------------------|
> | Static SNN | 1    | 95.01    | 9.6%             |
> | Static SNN | 2    | 95.64    | 30.0%            |
> | Static SNN | 3    | 96.26    | 57.3%            |
> | Static SNN | 4    | 96.46    | 100%             |
> | SEENN-I    | 1.09 | 96.07    | 11.8%            |
> | SEENN-I    | 1.20 | 96.38    | 13.8%            |
>
> Q3: The ablation study is not clear, it seems the SEEN adopts different backbones with SNN.
>
> A3: We apologize for any confusion in the ablation study. But we do use identical backbones when comparing SNNs and SEENNs. Therefore, the acceleration effect brought by our proposed method and the comparison is fair and clear. We will change our description and clarify it in the revised version of the manuscript.
>
> *Reference*
>
> [1] Kim, Youngeun, et al. "Neural architecture search for spiking neural networks." European Conference on Computer Vision. Cham: Springer Nature Switzerland, 2022.
>
> [2] Moitra, Abhishek, et al. "Spikesim: An end-to-end compute-in-memory hardware evaluation tool for benchmarking spiking neural networks." IEEE Transactions on Computer-Aided Design of Integrated Circuits and Systems (2023).

---

> > ### Author Response · Authors · 2023-08-18
> > **Authors reply**
> >
> > Thank you again for your time in reviewing our article. Given that the deadline for the author-reviewers discussion is approaching, we want to kindly remind you to let us know if you have further questions or concerns about our rebuttal. We welcome any further feedback, comments, or questions and we are open to discussion.

---

### Author Rebuttal · Authors · 2023-08-09

We’d like to thank all reviewers for their constructive feedback and suggestions on our work. We will address each reviewer’s questions and concerns point-to-point. And we welcome any further discussion on our paper.  We have attached a PDF file for demonstrating 4 figures. A detailed description can be found in each rebuttal thread.

Here, we'd like to reply to a concern that we did not include limitations and potential negative social impact. Our work focuses on reducing the inference cost of SNNs that can potentially benefit their deployment on edge hardware. We think it does not bring potential negative impacts on society. For limitations, SEENN uses hyper-parameter $\alpha, \beta$ to control the accuracy-efficiency tradeoff, which is less straightforward to users and may require certain trial and error.

---

### Decision · Program_Chairs · 2023-09-21

**Decision:**

Accept (poster)

**Comment:**

This paper received all four positive review scores. In the first round, reviewers raised some critical questions about the method's novelty, energy consumption, and experimental analyses. The authors provided detailed one-to-one responses to answer these questions. After the rebuttal and discussion, reviewers think the concerns are addressed and the opinions are converged to accept. The authors are encouraged to include the rebuttal materials in the revision, and the final recommendation is Accept.